# Charge Transport across Proteins inside Proteins: Tunneling across Encapsulin Protein Cages and the Effect of Cargo Proteins

**DOI:** 10.3390/biom13010174

**Published:** 2023-01-13

**Authors:** Riccardo Zinelli, Saurabh Soni, Jeroen J. L. M. Cornelissen, Sandra Michel-Souzy, Christian A. Nijhuis

**Affiliations:** 1Hybrid Materials for Opto-Electronics Group, Department of Molecules and Materials, MESA+ Institute for Nanotechnology, Molecules Center and Center for Brain-Inspired Nano Systems, Faculty of Science and Technology, University of Twente, P.O. Box 2017, 7500 AE Enschede, The Netherlands; 2Biomolecular NanoTechnology, Department of Molecules and Materials, MESA+ Institute for Nanotechnology, Molecules Center and Center for Brain-Inspired Nano Systems, Faculty of Science and Technology, University of Twente, P.O. Box 2017, 7500 AE Enschede, The Netherlands

**Keywords:** encapsulin, EGaIn, charge transport, biochemistry, molecular electronics, self-assembled monolayers

## Abstract

Charge transport across proteins can be surprisingly efficient over long distances—so-called long-range tunneling—but it is still unclear as to why and under which conditions (e.g., presence of co-factors, type of cargo) the long-range tunneling regime can be accessed. This paper describes molecular tunneling junctions based on an encapsulin (Enc), which is a large protein cage with a diameter of 24 nm that can be loaded with various types of (small) proteins, also referred to as “cargo”. We demonstrate with dynamic light scattering, transmission electron microscopy, and atomic force microscopy that Enc, with and without cargo, can be made stable in solution and immobilized on metal electrodes without aggregation. We investigated the electronic properties of Enc in EGaIn-based tunnel junctions (EGaIn = eutectic alloy of Ga and In that is widely used to contact (bio)molecular monolayers) by measuring the current density for a large range of applied bias of ±2.5 V. The encapsulated cargo has an important effect on the electrical properties of the junctions. The measured current densities are higher for junctions with Enc loaded with redox-active cargo (ferritin-like protein) than those junctions without cargo or redox-inactive cargo (green fluorescent protein). These findings open the door to charge transport studies across complex biomolecular hierarchical structures.

## 1. Introduction

Bioelectronics aims to utilize the convergence of nature-inspired biological architectures with electronics for applications ranging from stretchable and smart wearable electronics sensing to personalized healthcare [1,2,3,4,5,6,7,8]. For biomedical applications, soft, deformable interfaces between the device and human body are needed, or biosensors that accurately measure electrophysiological signals [9,10]. For example, nature-inspired devices that mimic squid-inspired fast color changing skin (constructed from nanoparticle materials inspired by *Doryteuthis pealleii* chromatophores [3]) have been used to develop soft robotics [3,11], foldable displays, electronic textiles, and color-changing devices. Besides applications, biomolecular devices are intriguing to study the mechanisms of charge transport (CT) [12,13,14,15,16]. In this regard, biomolecular tunneling junctions comprised of biomolecules sandwiched between two electrodes are of interest because they make it possible to study CT in the solid state [17,18,19,20], as opposed to electrochemical approaches that make it possible to study charge transfer in solution [21,22,23]. A variety of biomolecules have been used to construct tunneling junctions [12,19,24,25], including ferritin [15,26,27], bacteriorhodopsin [13,28], photosystem I [29,30], multi-heme cytochromes [31,32], DNA [6,24,33,34,35], and azurin [36,37], to study their rich behavior, including long-range tunneling, temperature-dependent and -independent CT [14], or current rectification [38]. The factors that affect the mechanism of CT are still unclear. For this reason, it is important to study how structural order (which may be different from the structural order in solution under physiological conditions [24,26,27]) defects in the junctions (roughness of the electrodes or effective contact area [19]), co-factors [5,16], orientation of the biomolecule [29], and size variation affect CT rates and their temperature dependency in tunneling junctions [15,20,24,29]. For instance, it is still unclear when the mechanism of CT is temperature dependent for which long-range tunneling [34] (temperature-independent CT over large distances routinely exceeding >10 nm [39]) is perhaps the most peculiar observation that still cannot be explained [5,12,39]. For all of these reasons, it is important to develop platforms that make it possible to systematically study the mechanism of CT.

Here, we report a new type of junction comprised of a very large protein-based cage, encapsulin (Enc), that readily encapsulates other proteins as its cargo, making it possible to systemically study CT as a function of cargo. Besides other approaches that allow us to study CT as a function of a co-factor, or protein cages of which the metal oxide content can be varied (ferritin and E2 [20,26,27]), Enc enriches the available toolbox to study biological CT in solid-state tunnel junctions by making it possible to study the CT of proteins inside a protein. Encapsulins are a group of protein cages formed by the assembly of a defined number of monomeric structures (Figure 1a) to give a hollow protein complex (Figure 1b) able to encapsulate cargo (as shown in Figure 1c,d) [40,41,42,43]. Here, we used Enc from *Thermotoga maritima* [44,45,46], which is a hyperthermophile bacterium, where the Enc is formed by 60 subunits (Figure 1a) to form a cage of 24 nm in the outer diameter with a cavity of 17–20 nm in diameter (Figure 1b). This cage encapsulates natural cargo, specifically a ferritin-like protein (FLP), whose monomer is shown in Figure 1c; this FLP can oxidize and store iron [42]. This is in sharp contrast to ferritin, where 24 subunits assemble into a cage that oxidizes and stores the iron inside the protein nanocompartment. In other words, Enc-FLP is composed of two proteins (Figure 1d): (i) the encapsulin assembly, which is made from 60 identical subunits responsible for the formation of the nanocompartment, and (ii) FLP contained inside the Enc nanocage responsible for the oxidation and storage of the iron ions [42]. The cargo of Enc possesses a cargo loading peptide (CLP) in C-terminal, which allows the encapsulation of the natural cargo but also of foreign cargo by adding this CLP in the C-terminal position [47]. The Enc used in our study was produced with two different purification tags: 10 histidine residues inserted in the position 127–128 (His-tag) [40] or an 8-residue specific for streptavidin (Strep-tagII). The protein cages were produced and purified from bacteria cultures along with different cargo. The cages inside the junctions had no cargo (Figure 1b), native FLP (Figure 1d), or super-folder green fluorescent protein (sfGFP, Figure 1e,f) as cargo [41]. Equus ferus caballus spleen apoferritin (HS-apoferritin), shown in Figure 1g, was used during the study as an internal reference. To understand the behavior of Enc and the effect of the cargo on the CT junctions of the protein, variants were prepared through chemical bond to a lipoic acid monolayer and by contacting them with an EGaIn electrode (EGaIn = eutectic gallium–indium), as shown in Figure 1h.

The production and characterization of encapsulin protein nanocages was analyzed via dynamic light scattering and transmission electron microscopy, followed by investigation of self-assembled monolayers (SAMs) on an Au surface via atomic force microscopy and current-voltage spectroscopy. We investigated tunneling CT through biomolecular tunneling junctions, as schematically shown in Figure 1g, using the EGaIn technique to form non-invasive and soft electrical contacts to different SAMs [48,49]. We showed that the Enc nanocages can act as encapsulating cages for separately investigating CT through redox-active (FLP) and neutral (sfGFP) cargos, wherein the former shows promising hysteretic and current rectification behavior. This opens new research avenues for studying CT in hierarchical (bio)molecular assemblies in solid-state devices consisting of proteins inside protein cages. 

**Figure 1 biomolecules-13-00174-f001:**
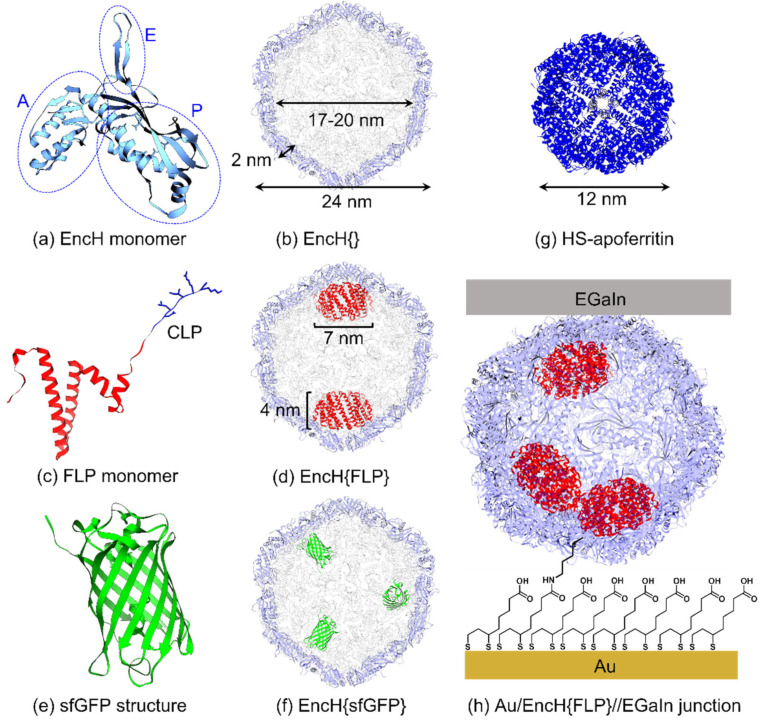
(**a**) Schematic of Enc monomer with highlighted axial domain (A) responsible for the generation of a pore with 5-fold symmetry, peripheral domain (P) that contacts a cargo loading peptide (CLP) through hydrophobic interactions, and elongated loop (E) that stabilizes the dimer. (**b**) Cross-section of EncH{} showing the encapsulin cage dimensions. (**c**) Ferritin-like protein (FLP) monomer with the main structure of the protein (in red) that is responsible for the decamer assembly and redox activity, and the CLP sequence (in blue). (**d**) Cross-section of EncH{FLP} showing 2 FLP cargo and their dimensions. The number of FLPs inside Enc can vary between 1–5 and the FLPs can be located in various positions (see TEM images below). (**e**) sfGFP protein β-barrel structure. (**f**) EncH{sfGFP} with 3 sfGFPs as cargo in the nanocompartment (though the numbers of sfGFPs and their positions can vary). (**g**) Equus ferus caballus spleen apoferritin cage structure. (**h**) Illustration of a complete EncH{FLP} tunneling junction with Enc immobilized on Au through a lipoic acid linker SAM and EGaIn top contact of the form Au/SAM//EGaIn (where ‘/’ and ‘//’ represent covalent and van der Waals interactions, respectively). The EGaIn surface is coated with a thin, few nm-thick GaO_x_ layer that provides stability to the junction [50,51]. All the protein structure schemes are made with Chimera software [52] based on the structure of Tm encapsulin (PDB 3DKT) [42].

## 2. Materials and Methods

The horse spleen ferritin, lipoic acid (thioctic acid), N-hydroxysuccinimide (NHS), 1-Ethyl-3-(3-dimethylaminopropyl)carbodiimide (EDC), EGaIn, and 95% extra-dry absolute ethanol was purchased from Sigma-Aldrich. Norland-61 optical adhesive was purchased from Norland Products Inc. and silicon wafers were obtained from the Okmetic company.

### 2.1. Protein Production and Purification

#### 2.1.1. Cloning and Production

The plasmid purification kit, restriction enzyme, and DNA polymerase were purchased from New England Biolabs (NEB). The polymerase chain reaction (PCR) amplifications to obtain the different genes of interest were conducted with high fidelity polymerase Q5 (obtained from NEB). The oligonucleotides used for the PCR were purchased from Eurofins Genomics and are listed in Table 1. The *enc* gene or the gene of the FLP cargo were amplified and respectively cloned into pRSFDuet and pETDuet vectors (Novagen) using the sequence and ligation-independent cloning method [53]. The *enc* gene was cloned between NdeI and EcoRV restriction sites and the *flp* gene between NcoI and SalI. The sequences of the obtained plasmids were analyzed by sequencing with the service of Eurofins Genomics.

Competent cells of *E. coli Rosetta* (DE3) strain were transformed with the appropriate plasmid for Enc or cargo, as summarized in Table 2. After an overnight preculture at 37 °C in Lennox Broth (LB) medium with appropriate antibiotics (streptomycin 30 μg/mL, kanamycin 30 μg/mL, and/or ampicillin 50 μg/mL), the bacteria were inoculated in 1 L of LB-antibiotic at a starting optical density at 600 nm (OD_600 nm_) of 0.1. They were grown at 37 °C until the culture reached an OD_600 nm_ between 0.4 and 0.6 and the expression of the genes was induced by adding 100 µM IPTG overnight at 25 °C. The bacteria were then collected by centrifugation for 10 min at 4000× *g* (further described in Appendix A).

#### 2.1.2. Purification

To purify (also described in Appendix A) the proteins, a pellet of bacteria was resuspended in lysis buffer (50 mM Hepes buffer pH 8, 150 mM NaCl, 1 mM (ethylenediaminetetraacetic acid) EDTA, 20 mM MgCl_2_, 1 mM phenylmethylsulfonyl fluoride (PMSF), 0.5 mg/mL lysozyme, 20 μg/mL DNAse, and 15 mM beta-mercaptoethanol (βme) supplemented either with 30 mM imidazole to elute protein with Histidine tag or 5 mM desthiobiotin to elute proteins with Strep-tagII. Bacteria were lysed by sonication (2 times for 1 min each) and the lysate was cleared with a first centrifugation at 4000× *g*. The supernatant was then cleared from membrane debris by ultracentrifugation at 140,000× *g*. 

The cleared lysate was purified by immobilized metal affinity chromatography (IMAC) and was then loaded either on a HisTrap HP 5 mL column (Cytiva, Marlborough, Mass, USA) when the protein was tagged with a His-tag or on a StrepTrap HP 5 mL column (Cytiva, Marlborough, Mass, USA) when it was tagged with a Strep-tagII, using a BioRad NGC FPLC (fast performance liquid chromatography). The unbound proteins were removed by washing with buffer (50 mM Hepes pH 8, 150 mM NaCl, 15 mM βme, 30 mM imidazole [His-tag] or 50 mM Hepes pH 8, 150 mM NaCl, 15 mM βme [Strep-tagII]). The encapsulin variants were eluted with elution buffer (50 mM Hepes pH 8, 150 mM NaCl, 15 mM βme, 500 mM imidazole [His-tag] or 50 mM Hepes pH 8, 150 mM NaCl, 15 mM βme, 2.5 mM desthiobiotin [Strep-tagII]).

We used size exclusion chromatography to purify the most concentrated fraction as follows. The elution was loaded on a Superose^®^ 6 Increase 10/300 GL column that was pre-equilibrated with a storage buffer (50 mM Hepes pH 8, 150 mM NaCl, 15 mM βme) to purify the assembled cages from aggregates and partially assembled encapsulin.

### 2.2. Sodium Dodecyl Sulfate PolyAcrylamide Gel Electrophoresis (SDS-PAGE)

The profile of the different purification steps was analyzed on SDS-PAGE with 15% acrylamide. The gels were stained with Coomassie brilliant blue G250. 

### 2.3. Dynamic Light Scattering (DLS)

The size distribution of the cages was determined by DLS with a Zetasizer (Malvern Panalytical, Malvern, UK). The sample was measured by 90° side-scattering and we determined the average of the intensity distribution from five experiments to determine the size. 

### 2.4. Transmission Electron Microscopy (TEM)

We applied 5 μL of protein sample to a Formvar carbon-coated copper grid (Electron Microscopy Sciences, Hatfield, UK). The samples were incubated on the grid for 2 min, with any excess buffer removed by wicking action by gently touching the droplet close to the sample side with a filter paper. The samples were negatively stained by applying 5 μL of uranyl acetate (1% *w/v*) onto the grid and incubating for 45 s. The excess was removed by filter paper. The samples were left to dry for 10 min before imaging on a Spectra300 microscope (Thermo Scientific, Waltham, Mass, United States).

### 2.5. Template-Stripped Au

An amount of 200 nm Au was deposited via e-beam vapor deposition on top of a monocrystalline silicon wafer (Si100). Glass strips (1.5 × 1.5 cm) were washed and sonicated with demi water, acetone, and isopropanol, and then glued onto the Au using a drop of Norland 61 optical adhesive. The glue was cured for 30 min under a UV light source (Bluepoint2 EasyCure Honle UV Technology) and then left to cure for 2 h in ambient conditions before use. The glued glass was lifted off the template with a razor blade revealing the ultra-flat, template-stripped gold surface and immediately (within a few seconds) transferred to the required solution for SAM formation. These samples were used for the AFM and EGaIn measurements described next. 

### 2.6. AFM Measurements

The sample was glued onto a magnetic holder and loaded onto the piezo-scanner of the AFM instrument (Bruker Multimode 8, by Bruker Corporation, Billerica, MA, USA). We used Bruker ScanAsyst-Air silicon tips on nitride levers (k = 0.4 N/m, r = 2 nm, and f_0_ = 70 kHz). A force of 1–2 nN was applied and the scan frequency was set to ~0.5 Hz. We collected 512 samples per line for 1 × 1 µm scans and 256 samples per line for the 250 × 250 nm scans. The roughness and height profile analysis was performed using Nanoscope Analysis software from Bruker.

### 2.7. Current-Voltage Measurements

We formed cone-shaped tips of eutectic gallium–indium alloy (EGaIn) and used those to form electrical contact to the monolayers using established procedures [54,55]. We made at least 15 junctions for each type of monolayer using more than 3–5 samples (3–5 junctions per sample). Each tip was used for 3–5 junctions. The EGaIn tip was biased and the gold surfaces were grounded. For each junction, 20 current density–voltage scans, *J*(V), were measured per going from 0 V → 1.5 V (or 2.5 V) → 0 V → −1.5 V (or −2.5 V) → 0 V, where steps of 0.05 V and 0.1 V were used for the ±1.5 V and the ±2.5 V sweeps, respectively. The measurements were interrupted only in case the junction shorted or became unstable. The data were analyzed using scientific Python and the publicly available code Gaussfit [56]. We determined the Gaussian mean value of log|*J*| for each applied *V* to construct the <log|*J*|>_G_ vs. *V* curves shown in the main text. From the data, we determined the rectification ratio and plotted the normalized differential conductance. The error bars represent the 95% confidence intervals. Further details of these procedures have been previously reported (see also Appendix A) [17,54,55,57].

## 3. Results and Discussion

### 3.1. Experimental Design

We aim to demonstrate that protein-inside-proteins structures are an interesting approach to design a new class of biomolecular junctions shown in Figure 1. Figure 2 (and Appendix A) shows a schematic of SAM formation of these different types of proteins which were then contacted with EGaIn electrodes to complete the junctions. To immobilize proteins on surfaces, we place freshly template-stripped Au surfaces in 3 mM degassed ethanolic solution of lipoic acid (C_7_S_2_COOH) for 2 h under an inert atmosphere similar to reported procedures [15,20,27]. Next, the samples were taken out of the solution and rinsed with degassed absolute ethanol (with ~2 mL). A degassed ethanolic solution containing EDC (75 mM) and NHS (50 mM) was prepared inside a glovebox followed by immersion of the lipoic acid SAM for 2 h to activate the linker SAMs (denoted as EDC-NHS) for protein immobilization. The substrates were then rinsed with absolute ethanol (~2 mL). A 0.10 mg/mL protein in buffer (Hepes 50 mM, NaCl 150 mM, pH 8) solution was prepared in ambient conditions. The activated substrates were immersed in the protein solution overnight at 4 °C and then rinsed with fresh buffer, milliQ water, and absolute ethanol (~1 mL each). The protein binds to several activated lipoic acid groups on the substrate via the lysine residues on the protein surface, and unreacted EDC-NHS hydrolyzes back to COOH groups in the presence of water in the buffer solution [58]. The SAMs were dried gently under a stream of nitrogen. 

We immobilized EncH with or without cargo: (i) Enc without cargo (EncH{}, Figure 1e); (ii) Enc with sfGFP protein cargo and with two different purification tags (EncH{sfGFP} and EncS{sfGFP} with 127H_10_ or His-tag and C-strep or Strep-tagII, respectively, shown in Figure 1f); and (iii) Enc with FLP as cargo (EncH{FLP}, Figure 1d,g). We characterized these biomolecular junctions by measuring statistically large number of current density–voltage, *J*(V), curves from which we determined the Gaussian log-average current densities, i.e., <log|*J*|>_G_ vs. *V* curves at 95% confidence levels using established methods [17,54,55,57]. By using Enc with different types of cargo proteins, we demonstrate that protein-inside-proteins structures allow us to control the CT properties of the junctions because Enc and GFP are redox-inactive in contrast to FLP which is redox-active (since its biological function is to oxidize and store iron oxide). This redox activity usually enhances tunneling rates and may also lead to electronic function (such as rectification of currents or memory effects) [14,17,29,38,59]. 

Protein Purification and Characterization

Figure 3 shows a typical size exclusion chromatogram for EncH, with the size exclusion chromatograms for the other variants given in Appendix A. Size exclusion chromatography showed that all four variants had the expected elution volume at 12.5 mL, in agreement with the literature for an assembled cage of 24–25 nm with a Superose^®^ 6 Increase 10/300 GL column [40]. The protein cages were also characterized in solution by using dynamic light scattering (DLS). Figure 3b show the DLS graph of EncH{} (Appendix A shows further results). This measurement shows a narrow distribution of the cages with a hydrodynamic diameter of 27–28 nm. Such a narrow distribution indicates that there is only one type of cage present in solution with respect of the purification tag and the cargo chosen (Figure 3b and Appendix A). Analysis with SDS-PAGE of EncH{FLP} and EncH{sfGFP} confirms the molecular weight of the monomer of EncH{} of 30 kDa and the purity of the monomer since only a few very faint other bands are visible (see Appendix A). 

The protein cages were further characterized with transmission electron microscopy (TEM) by drop casting the respective solutions of the proteins on a TEM grid followed by staining. Figure 4 show the images that were acquired with an electron beam intensity of 300 kV. These images show a homogenous distribution of the nanocages on the TEM grids in agreement with the size exclusion chromatograms, DLS, and SDS-PAGE results. Figure 4a was acquired with a 55,000× magnification and shows empty nanocages visible as bright “rings” because of the unstained peptides and dark core (filled with contrast agent). Figure 4b was acquired with a 55,000× and Figure 4c with 145,000× magnification revealing bright spots inside the nanocages of EncH{FLP}, indicating the presence of FLP inside the EncH cage. These experiments confirm the encapsulation of FLP cargo inside the nanocages and serve as proof-of-concept illustrating the stability of these nanocages and their encapsulated cargo outside buffer media (in this case, ultra-high vacuum).

### 3.2. Surface Characterization of EncH Self-Assembled Monolayers

We used atomic force microscopy (AFM) to image the surface morphologies of bare Au surfaces, the linker SAM, and the Enc-functionalized SAMs. Figure 5a (also Appendix A) shows the AFM images of template-stripped Au surfaces revealing a polycrystalline morphology with root-mean-squared (rms) surface roughness of 0.25 nm determined over an area of 1 × 1 μm^2^ (shown in Appendix A). From these results, we conclude that our template-stripped Au surfaces are ultra-flat in agreement with previously reported work [60,61]. After formation of the C_7_S_2_COOH linker SAM followed by activation with EDC-NHS, the rms surface roughness increased slightly from 0.28 nm to 0.75 nm, as tabulated in Table 3, suggesting that the SAMs are densely packed. Further, compared to the linker SAMs (C_7_S_2_COOH in Figure 5b and EDC-NHS in Appendix A), the AFM images with SAMs with Enc show visibly clear spherical shapes that are densely packed on the surface with a diameter of 25–30 nm estimated from the line scans (shown in Figure 5c–e). The height profiles further suggest that EncH{} and EncH{sfGFP} are flatter and have similar heights, but EncH{FLP} (due to its larger cargo size) has higher height variation. Additional wide-view 1 × 1 µm images of densely packed monolayers are shown in Appendix A. The native diameters of EncH{FLP} immobilized can be readily characterized by diluting the monolayer (by reducing the immersion time in EncH{FLP} buffer solution from overnight to 20 s), making it possible to image isolated proteins. Figure 4f shows an AFM image of individual EncH{FLP}, with the line scans revealing that the isolated proteins have a height of 7–10 nm and diameter of ~30 nm. These measurements suggest that the nanocages flatten on the surface, yet they remain intact. This result is expected since the protein assembly structure is plastic and depends on the solvation. As a consequence, upon dehydration compounded by the pressure exerted by the AFM tip during measurements, the proteins flatten because the nanocages are empty. These observations are similar to those obtained from other large protein cages on surfaces including monolayers of E2 [26]. 

### 3.3. Charge Transport Measurements

Figure 6 shows the <log|*J*|>_G_ vs. *V* curves recorded from junctions with the different types of Enc-functionalized SAMs (Figure 1) using the EGaIn technique as mentioned earlier. The data shown in Figure 6a were obtained by applying a voltage bias range of ±2.5 V, but additional data recorded using a smaller bias window of ±1.5 V are shown in Appendix A to demonstrate that the junctions were not altered by applying large bias. Figure 6c shows <log|*J*|>_G_ vs. *V* curves of junctions with SAMs of C_7_S_2_COOH linker, EDC-NHS linker, and HS-apoferritin as control experiments. Unsurprisingly, junctions with monolayers of Enc have substantially lower current densities than those junctions with the (activated) linker SAM because of the increase in tunneling barrier width of 7–10 nm due to the thickness of the Enc layer. Consequently, the junctions with Enc are very resistive in nature and relatively large bias of ±2.5 V is needed to see molecular effects. Indeed, the data shown in Appendix A are dominated by the capacitive nature of the junctions highlighting the resistive nature of these thick monolayers. Figure 6c also shows that the values of *J* for junctions with monolayers with activated EDC-NHS are lower than those junctions with the lipoic acid, which also can be explained by the notion that lipoic acid monolayers are thinner than EDC-NHS activated monolayers. To benchmark our work, we also measured the current density of junctions with HS-apoferritin monolayers (Figure 6c). The value of <log|*J*|>_G_ = −4.3 ± 0.5 A/cm^2^ at +1V is in agreement with the previously reported work on electronically similar ferritin cages lacking the His-tag [20,26], suggesting that the His-tag does not affect CT behavior in our junctions. These data also reveal that the junctions with Enc have approximately one order of magnitude higher resistance than those junctions with HS-apoferritin. The higher resistance of Enc is as expected, since HS-apoferritin junctions are of similar thickness as EncH{}, but the former is redox-active.

Junctions with monolayers of EncH{}, EncH{sfGFP}, and EncS{sfGFP} show very similar CT behavior, suggesting that the EncH{} cage is insulating in nature and the different purification tags, His-tag and Strep-tagII, do not affect the values of *J* even though His-tag is supposed to be more positively charged than Strep-tagII. Only a small rectification ratio is observed (Figure 6b), but, interestingly, these junctions show substantial hysteretic behavior with a difference in current in forward and reverse bias (rectification ratio, *R* = |*J*(−*V*)/*J*(+*V*)) of a factor of R < 10 at ±2.5 V. It is unlikely that this large hysteresis involves “trapped charges”, since these proteins are not redox-active but may be caused by movement of ions. It is likely that the cages are not completely empty and retain water (and perhaps salts from the buffer solutions); indeed, completely dehydrated proteins likely denature. For these reasons, we believe that trapped water and ions cause the observed hysteretic behavior [59].

Incorporation of redox-active cargo, i.e., FLP in EncH{FLP}, changes the CT rate and rectification behavior, as seen in Figure 6a,b. <Log|*J*|>_G_ increases from −1.98 ± 0.18 A/cm^2^ for EncH{} and to −0.22 ± 0.59 A/cm^2^ for EncH{FLP} at −2.5V. Figure 6b shows that at ±2.5 V, the current rectification ratio increases from R < 10 for junctions with EncS{sfGFP}, EncH{sfGFP}, and EncH{} to *R* = 100.48 for junctions with EncH{FLP}. These results show that the junction’s properties can be altered by changing the cargo of Enc. 

The effect of redox-active cargo in EncH{FLP} is also apparent for the ±1.5 V range in Appendix A, from the rectification and normalized differential conductance (*NDC* = |(d*J*/d*V*)*(*V*/*J*)|) plots, where a clear peak can be seen in the latter for *V* > 0. We hypothesize that the presence of the NDC peak and rectifying behavior in EncH{FLP} can be an effect of redox activity due to the iron-rich cargo which falls in our experimental bias window [57]. 

These findings of hysteretic and rectifying CT behavior in protein junctions with large barrier width require the development of better theoretical models for such large quaternary structure-based systems with host–guest geometries, different from the single-level CT models which are often used for protein-based tunneling junctions [25,31,37]. Such models will have to account for the macromolecular structure of these systems with multiple components affecting CT, instead of considering them as single molecular entities [32]. 

## 4. Conclusions

We introduce a new type of biomolecular tunnel junction based on protein-inside-protein architecture. Our results demonstrate a very useful functionality of these hierarchical biomolecular junctions where the overall properties of the junctions can be controlled by the cargo while the protein cage acts as an insulating container. The junctions are very stable and can withstand applied high voltages of 2.5 V, highlighting the efficacy of these SAMs as a robust platform for cargo-functionalized biomolecular devices. We included two types of cargo that was either redox-active (FLP) or redox-inactive (GFP), and found that the redox-active protein increased the tunneling rates by almost two orders of magnitude and the rectification ratio by one order of magnitude. This experiment showcases that it possible to tune the junction properties via the cargo. Interestingly, all junctions with Enc with and without cargo showed large hysteretic behavior, which is likely caused by trapped water and movement of ions. More detailed studies are needed to elucidate the mechanism of charge transport to establish if CT is dependent on temperature. Measurements as a function of relative humidity, different cargo proteins, and iron oxide loading of the cargo or thickness of the protein-inside-protein layers, combined with, for instance, impedance spectroscopy (especially at low frequencies) could shed more light on the CT mechanisms and on the role of water and ions. In the follow-up work, we aim to study CT with conductive probe techniques across individual proteins to establish how the number of cargo proteins change the CT properties of the junctions.

## Figures and Tables

**Figure 2 biomolecules-13-00174-f002:**
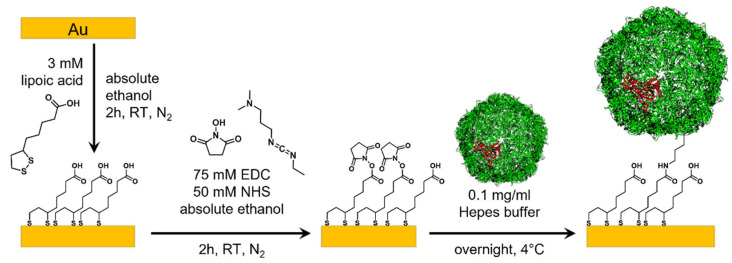
Schematic of encapsulin SAM formation. Au substrate is first functionalized using a 3 mM lipoic acid ethanolic solution in nitrogen atmosphere. After rinsing, most of the carboxylic groups of the lipoic acid are activated with an EDC-NHS ethanolic solution. Finally, the protein is chemically bonded onto the activated lipoic acid through the lysine residues. Note that the proteins may be linked to the surface with a varying number of linker SAM molecules and unreacted EDC-NHS hydrolyzes back to COOH groups in the presence of water in the buffer solution [58].

**Figure 3 biomolecules-13-00174-f003:**
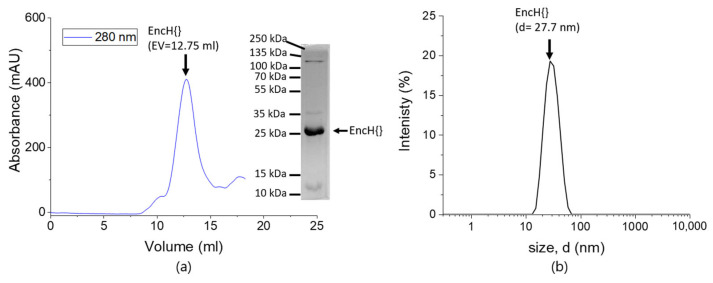
Purification results for EncH{}. (**a**) Size exclusion chromatogram of EncH{} showing the elution peak of the encapsulin cage centered at an elution volume of 12.75 mL. The inset shows the photo of SDS-PAGE with an intense band associated with the encapsulin monomer at 25 kDa. (**b**) DLS graph of EncH{} showing a monodisperse distribution of the protein centers at 27.7 nm, with no sign of disassembly or aggregation of the cage.

**Figure 4 biomolecules-13-00174-f004:**
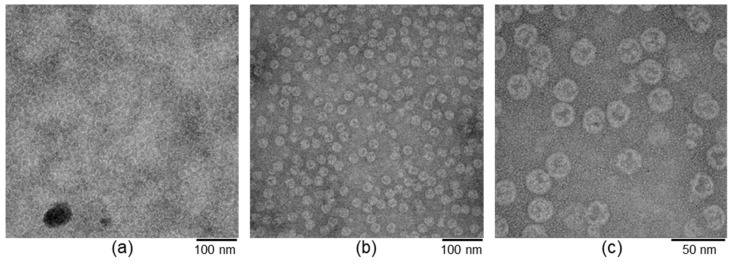
TEM images of drop-casted (**a**) EncH{} showing encapsulin cages as bright ring-shaped objects with a dark central area associated with an empty compartment, (**b**) EncH{FLP} showing most of the encapsulin cages filled with FLP in the nanocompartment, and (**c**) the same as (**b**) but with higher magnification.

**Figure 5 biomolecules-13-00174-f005:**
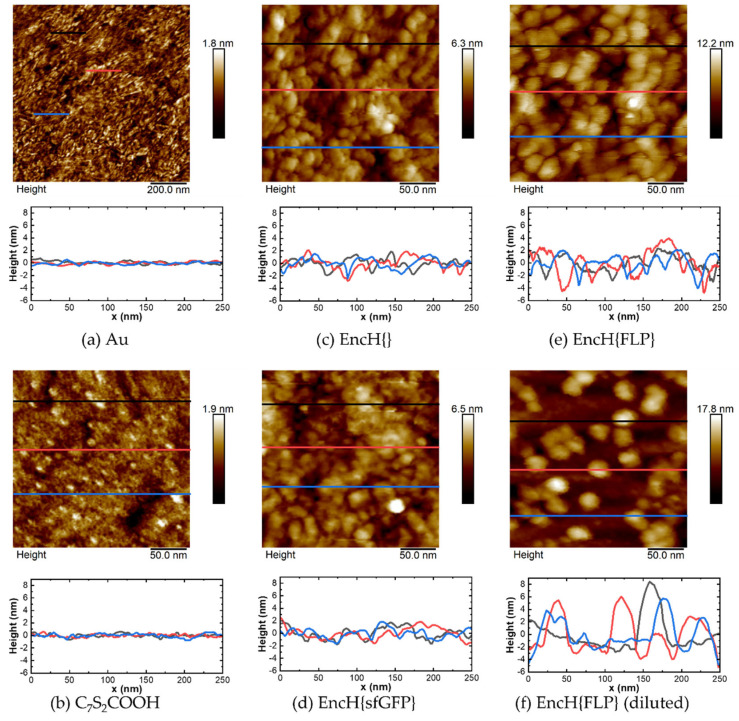
AFM images and height profiles of (**a**) bare Au surface and monolayers of (**b**) C_7_S_2_COOH, (**c**) EncH{}, (**d**) EncH{sfGFP}, and (**e**) EncH{FLP}. (**f**) AFM image of diluted SAM of EncH{FLP} showing individual proteins. The color of line in the AFM images corresponds to the color of the line plots in height profiles. Larger 1 × 1 µm images for the corresponding surfaces are shown in Appendix A.

**Figure 6 biomolecules-13-00174-f006:**
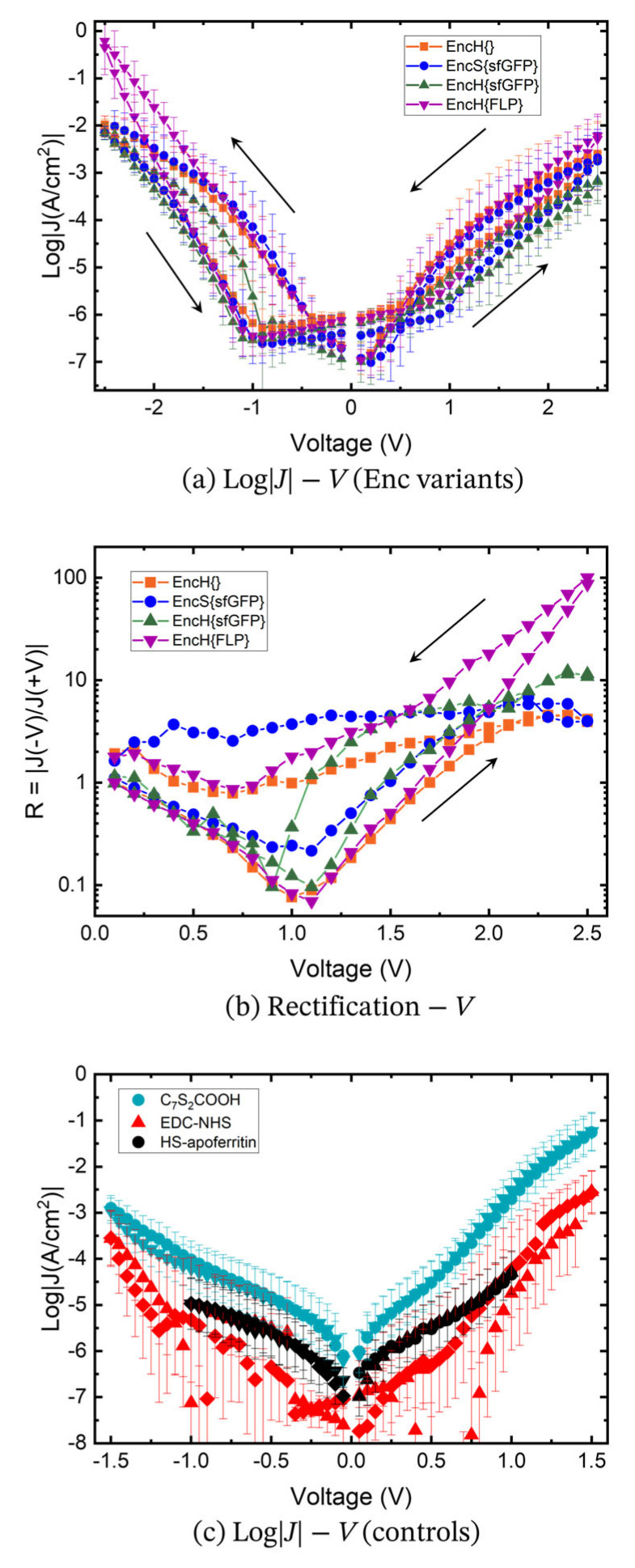
(**a**) Semilog plots of current density (log*|J*|) versus voltage (*V*) for the bias range of ±2.5 V and (**b**) plot of rectification *R* versus *V* for SAMs of EncH{} (orange), EncS{sfGFP} (blue), EncH{sfGFP} (green), and EncH{FLP} (purple). (**c**) Semilog plots of current density (log*|J*|) versus voltage (*V*) for the bias range of ±1.0 V for HS-apoferritin (black) and ±1.5 V for lipoic acid (cyan) and lipoic acid activated with EDC-NHS (red).

**Table 1 biomolecules-13-00174-t001:** Oligonucleotides used in this study.

Oligo	Sequence 5′-3′	Characteristics
OSMS3	AAGGAGATATACATATGGTGAACATGGAATTTCTG	PCR Tmenc forward for NdeI site in pDuet MCS2
OKB23	GCGTGGCCGGCCGATATCTCACTTTTCGAACTGCGGGTGGCTCCAACCACCACCACCACCGAACTTTAGAAGAATCAA	PCR Tmenc reverse for EcoRV site in pDuet MCS2
OSMS36	AGGAGATATACCATGGCAGATCAGTACCAC	PCR flp forward for NcoI site in pDuet MCS1
OSMS37	CCGCAAGCTTGTCGACTCAGAGCTTCCTTATG	PCR flp reverse for SalI site in pDuet MCS1

**Table 2 biomolecules-13-00174-t002:** Plasmids used in this study.

Plasmid	Characteristics	Origin
pCDFDuet-Tmenc127H_10_	Sm^R^, P_T7_, Expression of Tm encapsulin with a 10 Histidine tag after residue 127 (EncH)	Michel-Souzy et al. [40]
pRSFDuet-TmencCstrep	Km^R^, P_T7_, Expression of Tm encapsulin with a Strep-tagII after the C terminus (EncS)	this study
pETDuet-sfGFPE_flp_	Ap^R^, P_T7,_ Expression of the sfGFP fuserazmakd with the CLP of Flp (cargo)	Michel-Souzy et al. [40]
pETDuet-Flp	Ap^R^, P_T7,_ Ferredoxin-like protein (physiological cargo of Tm encapsulin)	this study

**Table 3 biomolecules-13-00174-t003:** Summary of all the EGaIn and AFM measurements performed for different SAMs in this project, including the number of junctions used in the statistical treatment and yields of non-shorting junctions for different bias ranges.

SAM	AFM Roughness ^a^		EGaIn Junction Parameters			
	250 × 250 nm	1 × 1 um	Voltage Range (V) ^c^	Non-Shorting Junctions ^c^	Total No. of Scans ^c^	Yield (%) ^c^
Bare Au	n.a.	0.25	n.a.	n.a.	n.a.	n.a.
C_7_S_2_COOH	0.25	0.28	1.5	22	440	76
EDC-NHS	n.a.	0.75	1.5	16	257	73
HP-apoferritin	n.a.	n.a.	1	17	340	74
EncH{}	0.89	1.19	1.5 (2.5)	15 (15)	278 (300)	79 (83)
EncH{sfGFP}	0.92	1.18	1.5 (2.5)	15 (15)	290 (300)	88 (100)
EncS{sfGFP}	n.a.	n.a.	1.5 (2.5)	19 (17)	380 (340)	73 (74)
EncH{FLP}	1.80 [2.72] ^b^	2.02 [3.08] ^b^	1.5 (2.5)	12 (18)	240 (360)	67 (90)

^a^ AFM roughness reported as root-mean-squared averages. ^b^ Roughness of diluted EncH{FLP} SAMs provided within square parentheses. ^c^ Values within parentheses are for ±2.5 V bias range, while the rest are for ±1.5 V bias range.

## Data Availability

Raw data are available on request.

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
