# Peer review of "Charge Transport across Proteins inside Proteins: Tunneling across Encapsulin Protein Cages and the Effect of Cargo Proteins"

_biomolecules, 2023, doi:10.3390/biom13010174_

Round 1

Reviewer 1 Report

The manuscript “Charge Transport across Proteins Inside Proteins: Tunneling across Encapsulin Protein Cages and the Effect of Cargo Proteins” by Zinelli et al. describes a new procedure testing protein transport properties. The manuscript focuses largely on the preparation and processing procedures, and provides some J-V characterization as proof-of-principle without much discussion of the observed transport behavior and mechanisms. Overall, I find the approach interesting, though it likely needs additional optimization before it can be utilized to fully understand protein electronic behavior.

In particular, the authors note that the Enc cages can encapsulate multiple copies of each of the chosen proteins, however, it is not clear whether they can or are separating the units by the number of proteins within the shell.  It appears that this should be possible given the weight differences that would occur with the number of proteins encapsulated.  The authors should discuss if this is achieved or achievable within the experiments.

An additional issue, both figures 1h and 2 show the final binding structure with the rest of the lipoic acid SAM with free carboxylic acids rather than with the NHS still present.  Is this correct? 

Finally, sometimes the authors refer to NHS-EDC, and other times to EDC-NHS.

Reviewer 2 Report

Report is attached
